# Recovering or Persisting: The Immunopathological Features of SARS-CoV-2 Infection in Children

**DOI:** 10.3390/jcm11154363

**Published:** 2022-07-27

**Authors:** Danilo Buonsenso, Piero Valentini, Cristina De Rose, Maria Tredicine, Maria del Carmen Pereyra Boza, Chiara Camponeschi, Rosa Morello, Giuseppe Zampino, Anna E. S. Brooks, Mario Rende, Francesco Ria, Maurizio Sanguinetti, Giovanni Delogu, Michela Sali, Gabriele Di Sante

**Affiliations:** 1Department of Laboratory and Infectivology Sciences, Fondazione Policlinico Universitario A. Gemelli IRCCS, 00168 Rome, Italy; maurizio.sanguinetti@policlinicogemelli.it (M.S.); michela.sali@policlinicogemelli.it (M.S.); 2Department of Woman and Child Health and Public Health, Fondazione Policlinico Universitario A. Gemelli IRCCS, 00168 Rome, Italy; piero.valentini@unicatt.it (P.V.); cristyderose@gmail.com (C.D.R.); rosa.morello91@gmail.com (R.M.); giuseppe.zampino@unicatt.it (G.Z.); 3Department of Translazional Medicine and Surgery, Section of General Pathology, Università Cattolica del Sacro Cuore, 00168 Rome, Italy; maria.tredicine@unicatt.it (M.T.); chiaracamponeschi94@gmail.com (C.C.); francesco.ria@unicatt.it (F.R.); gabriele.disante@unicatt.it (G.D.S.); 4Department of Biotecnologial, Basic, Intensivological and Perioperatory Sciences—Section of Microbiology, Università Cattolica del S Cuore, 00168 Rome, Italy; maria.perry@hotmail.it (M.d.C.P.B.); giovanni.delogu@unicatt.it (G.D.); 5Institute of Chemical Sciences and Technologies ‘‘Giulio Natta’’ (SCITEC)-CNR, Largo Francesco Vito 1, 00168 Rome, Italy; 6School of Biological Sciences, University of Auckland, Auckland 92019, New Zealand; a.brooks@auckland.ac.nz; 7Department of Medicine and Surgery, Section of Human, Clinical and Forensic Anatomy, University of Perugia, 60132 Perugia, Italy; mario.rende@unipg.it; 8Mater Olbia Hospital, 07026 Olbia, Italy

**Keywords:** pediatric Long COVID, childhood SARS-CoV-2 infection, COVID-19 immunopathology, T cells

## Abstract

Background. The profile of cellular immunological responses of children across the spectrum of COVID-19, ranging from acute SARS-CoV-2 infection to full recovery or Long COVID, has not yet been fully investigated. Methods. We examined and compared cytokines in sera and cell subsets in peripheral blood mononuclear cells (B and regulatory T lymphocytes) collected from four distinct groups of children, distributed as follows: younger than 18 years of age with either acute SARS-CoV-2 infection (*n* = 49); fully recovered from COVID-19 (*n* = 32); with persistent symptoms (Long COVID, *n* = 51); and healthy controls (*n* = 9). Results. In the later stages after SARS-CoV-2 infection, the cohorts of children, both with recovered and persistent symptoms, showed skewed T and B subsets, with remarkable differences when compared with children at the onset of the infection and with controls. The frequencies of IgD^+^CD27^−^ naïve B cells, IgD^+^IgM^+^ and CD27^−^IgM^+^CD38^dim^ B cells were higher in children with recent infection than in those with an older history of disease (*p* < 0.0001 for all); similarly, the total and natural Tregs compartments were more represented in children at onset when compared with Long COVID (*p* < 0.0001 and *p* = 0.0005, respectively). Despite the heterogeneity, partially due to age, sex and infection incidence, the susceptibility of certain children to develop persistent symptoms after infection appeared to be associated with the imbalance of the adaptive immune response. Following up and comparing recovered versus Long COVID patients, we analyzed the role of circulating naïve and switched B and regulatory T lymphocytes in counteracting the evolution of the symptomatology emerged, finding an interesting correlation between the amount and ability to reconstitute the natural Tregs component with the persistence of symptoms (linear regression, *p* = 0.0026). Conclusions. In this study, we suggest that children affected by Long COVID may have a compromised ability to switch from the innate to the adaptive immune response, as supported by our data showing a contraction of naïve and switched B cell compartment and an unstable balance of regulatory T lymphocytes occurring in these children. However, further prospective immunological studies are needed to better clarify which factors (epigenetic, diet, environment, etc.) are involved in the impairment of the immunological mechanisms in the Long COVID patients.

## 1. Introduction

Since December 2019, *SARS-CoV-2* (*CoV2*) has caused millions of infections and deaths, with a major clinical impact on fragile populations such as the elderly or those affected by other comorbidities.

Conversely, children have been relatively spared. Several pediatric studies from China, the US [1] and Europe [2] showed that most children develop a pauci-symptomatic or even asymptomatic infection. Severe and critical pediatric cases are significantly less frequent in children (respectively, 2.5% and 0.6%) than in adults [3,4,5,6,7], and deaths are extremely uncommon [8,9]. Age-specific differences in disease severity may be due to a lower susceptibility of children to infection, a lower propensity to showing clinical symptoms, different immune responses to the virus, or all of them [10]. A mathematical model based on epidemiological data from China, Italy, Japan, Singapore, Canada and South Korea estimated that the susceptibility to infection in subjects younger than 20 years might be half that of adults, with clinical symptoms manifesting from 21% of infections in 10- to 19-years-old, to 69% in individuals aged over 70 years [11]. However, this scenario is evolving as the number of pediatric cases is increasing due to low vaccine coverage in children and a higher transmissibility of the delta and omicron variants, although the overall clinical severity of pediatric COVID-19 during the omicron wave may be slightly reduced [12].

In a later phase of the pandemic, a rising number of patients’ organizations and researchers reported that patients were complaining of persistent symptoms after the resolution of acute infection. This condition, now known as Long COVID (or post-COVID condition—PCC), is characterized by symptoms such as fatigue, dyspnea, chest pain, cognitive and sleeping disturbances, arthralgia, and a decline in quality of life [13]. So far, there is not yet an internationally agreed definition of pediatric PCC. According to the NICE guidelines, the term “long COVID” is commonly used to describe signs and symptoms that continue or develop after acute COVID-19. It includes both ongoing symptomatic COVID-19 (from 4 to 12 weeks) and post-COVID-19 syndrome (12 weeks or more) (https://www.nice.org.uk/guidance/ng188/resources/covid19-rapid-guideline-managing-the-longterm-effects-of-covid19-pdf-51035515742 (accessed on 19 July 2022)). Conversely, a more recent Delphi process proposed a new definition of PCC, being the persistence of “at least one persisting physical symptom for a minimum duration of 12 weeks after initial testing that cannot be explained by an alternative diagnosis. The symptoms have an impact on everyday functioning, may continue or develop after *CoV2* infection, and may fluctuate or relapse over time [14]. For these reasons, data about its prevalence are affected by the definitions used and the included populations, ranging from 1% to 10% of cases, with older age and allergy being the main risk factors” [15]. Recently, cohorts of children with PCC have also been reported in Italy, Sweden, Russia, and the United Kingdom [16,17,18,19,20,21,22]. Many mechanisms have been proposed to explain PCC in adults, such as immune system dysregulation with a hyperinflammatory state, direct viral toxicity, endothelial damage, and microvascular injury [23,24]. While most of these works have been conducted in adults, a study has demonstrated that the same mechanisms may also be involved in children [25].

There is an urgent need to better characterize not only the protective and pathogenic immune responses during acute infections but also during the post-acute phases of *CoV2*. Several authors proposed a possible role of the “cytokine storm” in adults with COVID-19 [26], albeit others suggested that the clarification of the cellular responses may better explain the differences in the disease severity among adults and children [27,28]. Given the recent understanding that B and CD4^+^ T cells may in part help in understanding the differences in disease severity during acute infection [21,29,30], it is possible that similar mechanisms may also explain why some patients develop PCC.

To date, pediatric studies mainly focused on characterizing the immune responses during Multisystem Inflammatory Syndrome in Children (MIS-C), whereas an important gap exists in the study of the immunological responses during the different phases of infection (from acute infection to post-acute stages) in children affected by COVID-19. Therefore, we performed this study aiming to explore the immune responses in children presenting with the SARS-CoV-2 infection, covering the whole spectrum of disease, including acute infection, full recovery, or PCC, according to the NICE guidelines.

## 2. Materials and Methods

### 2.1. Study Population

This is a prospective study of children younger than 18 years of age with a microbiologically confirmed diagnosis of *CoV2* infection (based on *CoV2* detection through nasopharyngeal swab and its analysis by qRT-PCR), assessed in our institution, either in the Emergency Department or Pediatric ward due to acute disease, or in our pediatric post-COVID outpatient clinic. In our outpatient clinic, we evaluated children either fully recovered from acute infection or presenting with persistent symptoms. Children were referred to the post-COVID unit either after discharge from our institution or directly from the family pediatricians (and therefore not seen at the baseline during acute infection).

For our study, we enrolled the following three categories of children:− Children with acute *CoV2* infection (a microbiologically confirmed diagnosis through RT-PCR on nasopharyngeal swab by).− Children with PCC after microbiologically confirmed (with PCR on nasopharyngeal swab) acute COVID-19 were identified using an internationally developed survey (https://isaric.org/research/covid-19-clinical-research-resources/paediatric-follow-up (accessed on 4 July 2022)). Since there is not yet a consensus definition of post-acute sequelae of SARS-CoV-2 infection (PASC) in children, we defined as “PCC children” those having at least one persistent symptom for more than eight weeks after the diagnosis of acute COVID-19, according to recent studies in children and the NICE guidelines (https://www.nice.org.uk/guidance/ng188/resources/covid19-rapid-guideline-managing-the-longterm-effects-of-covid19-pdf-51035515742 (accessed on 19 July 2022)). Specifically, we chose the NICE guidelines because the persistence of symptoms for 8 weeks in children, who usually suffer from mild disease during acute infection, places an important burden on a child’s and a family’s quality of life, and patients seek medical attention before the 12 weeks cut-off.− Recovered children: those that reported no persistent symptoms after acute *CoV2* infection and that were assessed at least 28 days from the onset of COVID-19 symptoms. 

#### Inclusion and Exclusion Criteria

The following inclusion criteria were used:Children aged 0–18 years.The child sought/needed primary or secondary medical care for COVID-19.Laboratory (RT-PCR) diagnosis of acute COVID-19.>28 days from the onset of COVID-19 symptoms.Parent’s/carer’s/guardian’s consent to participate.

The following exclusion criteria were used:

Patients with confirmed or suspected primary or acquired immune compromising conditions, recent or current administration of immune suppressive therapies, or other diseases affecting the immune system, or patients with chronic comorbidities and genetic disorders that might affect immune responses or the recognition of subtle symptoms of Long COVID difficult. Additionally, children fulfilling WHO criteria for MIS-C were excluded, since recent studies suggest a specific immunological signature for this condition [31].

### 2.2. Immunological Studies

#### 2.2.1. Cytokines Analyses

The evaluation of the expression levels of different human cytokine (IL6, IL1β, TNFα and IL8) was performed on patients’ serum using the ELLA system (ProteinSimple, San Jose, CA, USA) according to the manufacturer’s protocol.

#### 2.2.2. Flow Cytometry Analysis

The direct labelling of blood was performed using pre-coated and dried antibodies combined in predesigned panels (DURAClone^®^ technology, [21,22]). Whole blood samples were stained using DURAClone IM Treg Tube and DURAClone IM B cells Tube [32]. CytoFLEX V5-B5-R3 Flow Cytometer and Kaluza Analysis 2.1 software (Beckman Coulter, Pasadena, CA, USA) were used for the analyses. Examples of a gate strategy for Regulatory T (Treg) and B cells analysis are reported in [21]. Regulatory T cells were gated partially following the manufacturer’s protocol [21]. We measured the frequency of CD4^+^CD25^high^FoxP3^+^ Tregs, subdividing them into CD45RA^−^Helios^+^ natural Tregs (nTregs), CD45RA^−^Helios^−^ inducible Tregs (iTregs) and CD45RA^−^Helios^−^CD39^+^ suppressor Tregs. The percentages have been obtained by the normalization as following: Tregs on the normalized CD45^+^CD4^+^ cells; iTregs, and nTregs on the normalized Tregs; and suppressor iTregs on the number of normalized iTregs. The manufacturer’s protocol was followed for the B-cell subsets gating strategy [21]. We measured the frequency of CD45^+^CD19^+^ B cells, subdividing them based on CD27/IgD expression (IgD^+^CD27^−^ naïve, IgD^−^CD27^+^memory, IgD^+^CD27^+^marginal zone and IgD^−^CD27^−^ double negative B cells) and based on IgD/IgM expression (switched and unswitched B cells). The above-mentioned subpopulations were normalized as percentages on CD45^+^CD19^+^. The IgD^−^IgM^−^CD27^hi^CD38^hi^ Plasmablasts were gated and normalized on IgM^−^IgD^−^ B cells percentages, while IgM^+^CD27^−^CD38^dim^ were gated and normalized on IgM^+^IgD^+^ B cells percentages. The CD38^+^CD24^+^ transitional B lymphocytes were gated and normalized on IgD^+^IgM^+^CD27^−^CD38^dim/high^ B cells percentages.

### 2.3. Ethic Committee Approval

This study has been approved by the ethics committee of Gemelli University hospital (ID 3078). Oral and written informed consent was obtained from children older than 14 years of age and from their guardians, or from the guardians only for patients younger than 14 years of age.

### 2.4. Quantification and Statistical Analysis

The GraphPad Prism 9.3.1 software was used for the analysis of the data. The figure and tables legends provide the statistical details of each experiment. Data plotted in linear scale were expressed as mean + standard deviation (SD). The effects and possible interaction(s) of independent variables were examined using two-way ANOVA corrected with Tukey. *p* values ≤ 0.05 were considered significant. Details pertaining to significance were also noted in the respective legends.

## 3. Results

### 3.1. Study Population

We enrolled 141 children: 49 *CoV2*^+^ children at the onset of the disease, 51 children affected by Long COVID, 32 children that recovered after *CoV2* infection, and 9 healthy children. Main demographic characteristics, COVID-19 severity and the main persistent symptoms are described in Table 1.

### 3.2. Perturbation of Circulating B Cells and T Regulatory Subsets by CoV2 Infection in the Childhood

Peripheral blood samples were collected from children at the onset of the infection (at the diagnosis), from patients accessing our Infectious Disease Pediatric Unit, to evaluate the recovery or persistence of symptoms, as described in the method section, and from non-infected controls. Circulating lymphocytes were analyzed for B and regulatory T (Tregs) subpopulations. The gating strategies are described in [21]. No significant differences were found in the white blood cells (WBC) count between the patients and controls in peripheral blood. As expected, the total amount of T and B lymphocytes were inversely proportional with the age of the individuals (data not shown). The percentages of B and Tregs subpopulations were gated on total amounts of CD19^+^/CD45^+^ and CD4^+^/CD45^+^ cells, respectively, aimed at normalizing the age-dependent differences. We normalized T and B-cell subsets on the total amount of circulating lymphocytes from a routinary blood test (Appendix A).

B-cell subpopulations in control group were consistent with those expected based on the reference ranges for sex and age [33,34]. Children at disease onset showed perturbed distributions of B-cell subsets at data entry (Figure 1A and Table 2); specifically, the frequencies of IgD^+^CD27^−^ naïve B cells, IgD^+^IgM^+^ cells and IgM^+^CD27^−^CD38^dim^ B cells were higher in children with recent infection than in those with an older history of disease, as detailed in Figure 1A. Circulating memory B cells (CD27^+^) did not change in the three groups of patients, while switched B-cell subsets appeared to be expanded in few children with persistent symptoms or who fully recovered from *CoV2* infection, highlighting the heterogeneity of these groups of patients. Interestingly, B-cell subsets did not show a different modulation among recovered and children with persistent symptoms.

We also analyzed the regulatory compartment of T helper lymphocytes, finding that CD25^high^CD127^low^FOXP3^+^ (Tregs) were expanded in children at the onset of disease compared to the controls and to the children who fully recovered from symptoms, as displayed in Figure 1B. Gating strategies with CD39, Helios and CD45RA allowed for the finer characterization of this subset, showing that this different distribution involved the natural Tregs more than the inducible subset, suggesting the peculiar mechanisms of immune regulation in children. However, the regulatory T cell subsets did not show a different modulation among children with persistent symptoms and the ones who fully recovered (Table 2, c column of *p* values).

### 3.3. Heterogeneous Expression of Inflammatory Cytokines in CoV2+ Children

Inflammatory cytokines were measured in the serum samples collected from children and stratified according to *CoV2* infection history (Figure 2). We focused on IL1β, TNFα, IL6 and IL8 concentrations. No remarkable differences between groups were detected, except for a subgroup of patients at disease onset who showed increased levels of all these cytokines, especially of IL8, significantly upregulated in all three groups compared with controls (ANOVA multiple comparison with Tukey correction, Figure 2B). Although we were not able to find a correlation between inflammatory cytokines and age, we found that asymptomatic *CoV2*+ children and age-matched controls showed comparable low levels of IL1β and IL6 (data not shown). Cytokine levels were not differently modulated between recovered children and children with persistent symptoms.

### 3.4. Age-Related Distribution of B Cells and Treg Subsets in CoV2+ Children

To dissect the potential bias introduced by the comparison of children of different ages at different stages of immunological development and to minimize the heterogeneous distribution of B cells and Treg subsets among patients of different ages, we selected 6 years as a possible cut-off point to distinguish two groups of patients/controls with different immunological behavior, excluding preschoolers (displayed in Appendix A). Interestingly, most of the significant differences shown in Figure 1 regarding the B cell compartment were detectable in the groups of children older than 6 years (Figure 3A and Table 3). Indeed, the impact of *CoV2*, during the early stages of the infection, is evident in immature B cells (naïve, unswitched and pre-transitional), which, conversely, seem to be contracted during the late phases. Comparing the recovering and persisting of symptoms in children older than 6 years, significant differences between naïve and unswitched B cell compartments were detected, suggesting a possible role of these subpopulations in sustaining the disease persistence.

Conversely, the older children showed unperturbed Tregs (Figure 3B and Table 3). Moreover, as expected, the natural Tregs, widely modulated by *CoV2* infection, as displayed in Figure 1B, were poorly represented in the children older than 6 years. The immune modulation by early/late phases of *CoV2* infection appeared to be determined also by the T regs subsets in preschooler children (<6 years old, Appendix A), while the persistence of symptoms seemed to be dependent on age-related B responses, although in none of the other subgroups was it possible to detect a difference in terms of memory B cells (Figure 3A and Figure Appendix A).

### 3.5. Sex-Related Modulation of Immune System in CoV2+ Children

To dissect whether the modulation of T and B lymphocyte subpopulations was sex-dependent, we re-evaluated female and male cellular immune response with respect to the distribution of B cells and Treg subsets (Figure 4A–D). B-cell-mediated responses appeared with similar modulations in the different groups of children, independently of sex (Figure 4A,C), with a comparable significant high amount of immature B cells (naïve, unswitched and pre-transitional) during the early phases of the disease. No significant differences were detectable among children with recovered or persistent symptoms, either in females or males. However, the only subtle difference concerned the fact that higher percentages of pre-transitional B cells can be found in the group of males during the early stages of infection than in the females (Figure 4A,C). B-cell subsets did not show a different modulation between recovered children and children with persistent symptoms.

Conversely, we found that *CoV2* infection causes the perturbation of regulatory T cells during childhood in a sex-dependent manner. The Tregs subsets in females were lower and were not differentially expressed among groups, except for natural Tregs (Figure 4B). The Tregs percentages correlated with disease activity in males and, in line with this, similar trends of the presence of total and inducible Tregs were detectable in both onset and Long COVID groups (Figure 4D).

### 3.6. Peripheral Blood Distribution of B Cells and Treg Subsets during Follow Up

The distribution of B cells and Treg subsets was analyzed in 24 children again after the initial enrolment of patients during a visit for the re-evaluation of the symptoms/wellness. Seventeen patients were fully recovered and negative for *CoV2* (molecular test), even if they started from different clinics at onset, 2 were asymptomatic and 15 presented with symptoms (6 mild, 4 moderate and 5 severe). These follow-ups were compared with 7 children with persistent symptoms both at study entry and at the point of the re-evaluation. Despite the limited numbers of these two groups, it was possible to note a different evolution of the Tregs and B-cell compartments as displayed in Figure 5. Peripheral B cells did not differ significantly among recovered and persistent groups (Figure 5A–D), although naïve and switched IgD^−^ B lymphocytes had a trend of reduction in children with persistent symptoms, while they were stable in patients who fully recovered (Figure 5B,C). In both groups, an increase of both peripheral memory and pre-transitional B cells was evident (Figure 5A,D). Interestingly, the regulatory T cell compartments display different trends: total Tregs increase in patients who fully recovered, while the group with persistent symptoms was not able to reconstitute the total Treg (Figure 5E). The slopes of total Tregs were not significantly different, probably limited by the low number of cases. Inducible and suppressor Tregs increased in both groups (Figure 5F,G), although the group with persistent symptoms was less efficient in terms of delta and the slope of the linear regression. However, natural Tregs were significantly different among the two groups, increasing in children who fully recovered from the *CoV2* infection (Figure 5H, F = 5.761. DFn = 1, DFd = 20 and *p* = 0.0262); patients with persistent symptoms presented higher levels and a reduction of peripheral natural Tregs, correlating this compartment to the persistence of the disease.

## 4. Discussion

In this study, we described the clinically and temporal effects of *CoV2* infection on the modulation of the immune profile of children at different stages, from onset and with different habits of the acute disease, to advanced phases comprising both recovery and Long COVID states. To our knowledge, this is the first immunological study that has included children with a clinical diagnosis of Long COVID. Overall, we have found that children with acute infection have a different profile compared with those in the post-acute phase. Regarding children with Long COVID, although we found a significant overlap with those that fully recovered from the infection, some of them showed a profile similar to the acute infection in terms of effector T and switched B lymphocytes, suggesting that some Long COVID children are not able to switch from an innate to an adaptive immune response.

Since the beginning of the pandemic, we have been aware that age is one of the main determinants of the clinical outcome of *CoV2* infection [35]. Several reports demonstrated an enhanced and cross-reactive humoral and cellular immune response to *CoV2* during childhood, both markedly skewed to spike more than to nucleocapsid and envelope proteins and with a more highly differentiated profile [36]. Starting from the different immune response in children and speculating that this enhanced and reactive immune response in children contributes to the excellent clinical outcomes of this group, compared to adults, we hypothesize that the immune profile could represent a potential biomarker of the case of pediatric Long COVID [37,38,39].

Although there are no other pediatric studies evaluating the long-term immune profile after acute infection, several studies have addressed this in adults, including those with Long COVID. This topic recently also became relevant in adults, although the mechanisms involved in the development of Long COVID symptoms are still debated and the reasons for the clinical persistency in a minority of patients are multifactorial and heterogenous [40]. Interestingly, Chansavath Phetsouphanh et al. found that, similarly to our patients, adults with Long COVID had highly activated innate immune cells, which is in line with a subgroup of our pediatric patients that appeared to fail to switch to an adaptative immune response [41]. However, adult Long COVID patients showed persistently high levels of pro-inflammatory cytokines, and, in particular, the combinations of the inflammatory mediators IFNβ, PTX3, IFNγ, IFNλ2/3 and IL6 were associated with Long COVID with 78.5–81.6% accuracy. Differently from these findings, in our cohorts during the whole spectrum of *CoV2* infection, we were not able to detect enhanced levels of pro-inflammatory cytokines associable to clinical outcomes. These differences, however, are in line with clinical observations that pediatric Long COVID is usually milder than that the adult form [42]. Therefore, it is not surprising that children exhibit a milder inflammatory background, as also happens during acute infection [41]. Of note, persistent T cell abnormalities were also found in an independent cohort of Irish patients during convalescence, three months or more after the initial *CoV2* infection, which were more marked with age and independent of ongoing subjective ill-health, fatigue, and reduced exercise tolerance [43]. In addition, Ryan et al. followed up on COVID-19 patients up to 24 weeks, finding markers of T and B cell activation/exhaustion [44], as also reported by others [45,46,47,48].

More specifically, one of the most interesting results of our study concerned the regulatory T cells. As shown in Figure 1B, the regulatory compartment of Long COVID children was more similar to those with early-stage acute infection than those that fully recovered from disease. This finding deserves attention since the expansion of naïve Tregs could be an attempt to restore the balance in the Treg pool in the face of both inflammation and tissue damage, which is supported by emerging evidence of a dual role for Tregs in suppressing immune responses and promoting tissue repair [49,50,51], while the persistently expanded effector T lymphocytes suggest that in some children the immune system is still activated by some persistent stimuli [52,53,54,55].

Interestingly, although analyzed in only a minority of patients, we also found that some children with persistent Long COVID, analyzed during different time points, have a declining number of total and natural Tregs (Figure 5E,H) and of naïve and switched B lymphocytes (Figure 5A,B) that produces IgM and IgG. These data further reinforce the hypothesis that some patients remain in an active immune activation status rather than switching to an adaptive post-acute phase. Although we have not performed extensive sub-phenotyping of the immune responses and we have not analyzed these signatures after specific stimulation with *CoV2* proteins, these findings are in line with adult data [44] but, more importantly, with a growing body of knowledge that Long COVID patients may present persistent viral antigens that may subtly stimulate the immune system [15]. In fact, persistent viral particles have been described in several tissues [15], including in pediatric reports, and some studies have also linked chronic, abnormal gastrointestinal stimulation of the immune system with the development of Multisystem Inflammatory Syndrome in Children (MIS-C) [56], another post-acute complication of acute *CoV2* infection in children.

We found a wide overlap in immune profile in our cohorts of recovered and Long COVID children, which suggests that other mechanisms and deeper immunologic investigations are needed to better characterize a biosignature of pediatric Long COVID, which is urgently needed since the clinical presentation is subtle and the condition is difficult to diagnose. In fact, it is also possible that some of our patients may have been misclassified since some symptoms are also common in children that tested negative to COVID-19 [57]. In any case, there is no doubt that other mechanisms can play a role. For example, Liu et al. found that convalescent patients have significant impairment in Natural Killer T cells [58], while Siska and colleagues found immunometabolic dysregulation in COVID-19, which persisted even after infection and was documented by higher levels of intracellular reactive oxygen species and disrupted mitochondrial architecture [59]. It is still unknown if this happens as well in *CoV2*-infected children, and how this might reflect in younger children that have more frequent exposure to common pathogens and viruses. More importantly, Su and colleagues found that, in adults, multiple early factors anticipate Long COVID, including the post-acute expansion of cytotoxic T cells (which was associated with gastrointestinal symptoms); subclinical autoantibodies; and the reactivation of latent viruses, specifically Epstein–Barr virus [60]. This last finding is specifically important since this has also been documented by other independent cohorts [61]; moreover, children and adolescents have high risks of recent exposure to EBV infection, a virus with known relationships with chronic fatigue syndrome/myalgia encephalomyelitis and other long-term consequences [15].

Regarding the B cell compartment, we have not found specific characteristics of interest in the post-acute phase. However, since we have not performed assessments after stimulation with *CoV2* proteins, this was not unexpected since in post-acute phase B, cells are stored in the bone marrow. It is important to highlight that specific pediatric populations, including those immune deficiencies, may present specific characteristics in post-acute immune responses, as demonstrated during acute infection or vaccination [62,63]. In fact, as we have excluded these populations, our results may not be directly translated in children with diseases affecting the immune system.

## 5. Limitations of the Study

Our study has limitations. The main limitation is the small number of children enrolled. However, the more severe spectrum of COVID-19 in children is relatively rare, and therefore only a limited number of children seek medical care in hospitals, while the majority of children are managed at home or in the outpatient settings. Similarly, Long COVID in children is less common than in adults, justifying the relatively low number of children with Long COVID that we included. Therefore, a multicenter immunological study is necessary to overcome this limitation. Secondly, we did not include cases of Multisystem Inflammatory Syndrome in Children (MIS-C) temporally related to *CoV2*. We decided not to include these patients because the immunological signatures of these patients have been studied more thoroughly than COVID-19 and Long COVID [31], and the current evidence suggests that MIS-C has a well-defined different immune pathogenesis compared with *CoV2* infection. Despite these limitations, our study represents the first attempt at providing a detailed immune profile of the spectrum of the different outcomes of *CoV2* infection in children, from acute infection to Long COVID. Last, an internationally recognized definition and diagnostic test of Long COVID, specific to children, is still lacking, leading to the possible misclassification of children within this group. At the time of the beginning this study, there was only an adult definition for Long COVID provided by the WHO (https://www.who.int/publications-detail-redirect/WHO-2019-nCoV-Post_COVID-19_condition-Clinical_case_definition-2021.1 (accessed 20 April 2022)); the first pediatric one was only recently achieved after a Delphi process [14]. However, the definition we used for our study is mostly similar to both the WHO and UK proposed ones. In any case, since both definitions are not specific and most also rely on the subjective reporting of symptoms by the patient/caregivers and the subjective interpretation of the healthcare professionals, it is possible that we have classified as Long COVID some children with fatigue of another origin, including psychological issues. This might explain the wide variation of the immune profile we have found in our cohort. Additionally, since we only included 51 children with Long COVID, we do not have enough power to make stratifications according to different clinical phenotypes; this would be an important next step in Long COVID research, since there is growing evidence that Long COVID mostly covers a spectrum of different clinical presentations (e.g., prevalent gastrointestinal symptoms vs. fatigue vs. neurocognitive, or single vs. multiple symptoms, further stratified by age and sex [47]), which may indeed be explained by different pathological/immunological mechanisms, or by the prevalence of some events over others (e.g., endothelial inflammation, viral persistence, and gut dysbiosis [24]), which translate into different immunological signatures. Importantly, not all children have been enrolled at the same time since acute infection. However, ours is the very first attempt of an immunological profile in children along the spectrum of *CoV2* infection including Long COVID, providing some findings that are coherent in relation to recent adult studies [64] and informing the development of future studies, supporting the claim that some unknown mechanisms can drive Long COVID in some children rather than being a simplistic psychological consequence of social restrictions [65]. Certainly, future studies performing more immune subsets and deeper immune profiling might reveal further clues in the immunopathology of PCC in children.

## 6. Conclusions

In conclusion, our study provides preliminary evidence that some children with Long COVID show an abnormal switch from innate to adaptive immune responses, documented by the low representation of effector T cells and of IgD- B cells. The drivers of this immune dysregulation require further investigation with new-generation immunological studies, and the exploration of other mechanisms is also needed, including autoimmunity, viral persistence, and chronic endothelial inflammation, with the goal to better understand, recognize and manage pediatric Long COVID.

## Figures and Tables

**Figure 1 jcm-11-04363-f001:**
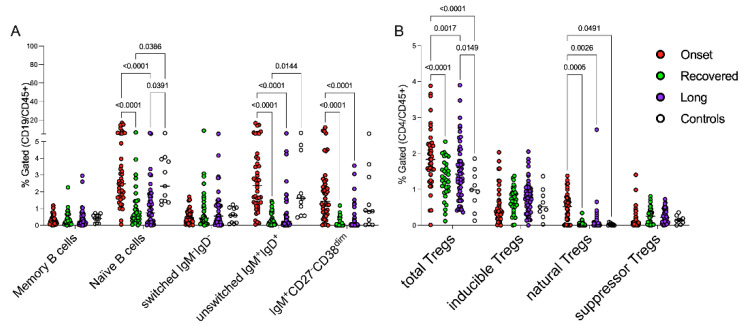
Peripheral blood B and Tregs subsets during the immunopathology of *CoV2* infection. Two different panels of antibodies for B cells and Tregs subsets (Duraclone^®^, Beckman Coulter, Pasadena, CA, USA) were used for staining. Gating strategy for the identification of cell subsets is described in [21]. Each circle represents the percentage of each cell population of the different patients/healthy subjects based on the indicated colors: *CoV2*-infected children at onset of the disease (red circles), who recovered (green circles), with persistent symptoms (violet circles) and healthy subjects (white circles). The panel (**A**) displays the different distribution of B-cell subpopulations; panel (**B**) shows the percentages of Tregs subsets in the different groups. Symbols–or + were used to identify subpopulations with positive or negative markers, respectively, while “high” or “dim” indicate highly or moderately expressed markers respectively. A two-way ANOVA corrected with Tukey were used to statistically examine the data.

**Figure 2 jcm-11-04363-f002:**
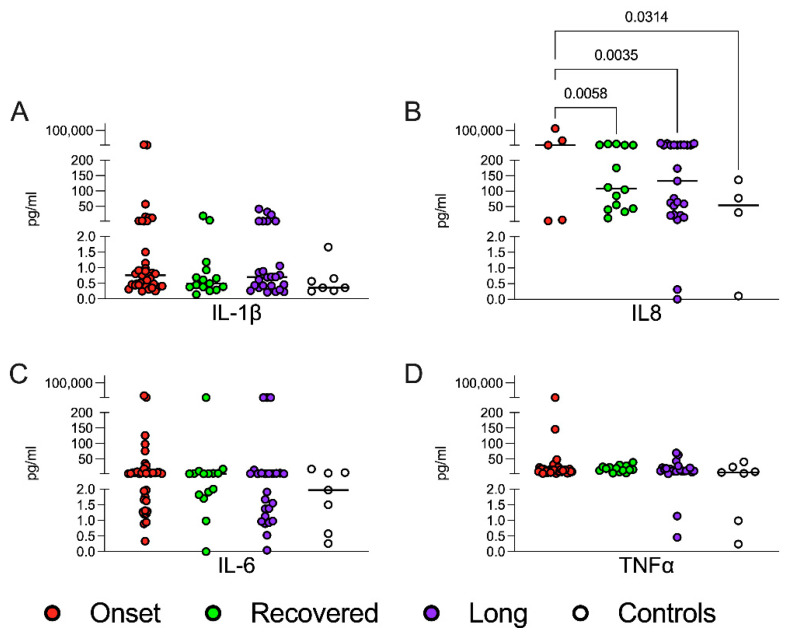
Cytokine levels in the sera. Cytokine expression was measured in sera samples of patients and healthy subjects at data entry (ELLA Assay, ProteinSimple, San Jose, CA, USA). Each circle represents the concentration (expressed as picograms/milliliter, pg/mL) of the cytokines for each patient/healthy subject, and the 4 plots compare the four groups. Statistical analyses were performed with a two-way ANOVA corrected with Tukey. (**A**) IL1β: interleukin 1 β; (**B**) IL8: interleukin 8; (**C**) IL6: Interleukin 6; (**D**) TNFα: tumor necrosis factor α. IL8 plot (Figure 2B) displays a limited number of patients/healthy subjects. The missing ones are not shown because they were undetectable.

**Figure 3 jcm-11-04363-f003:**
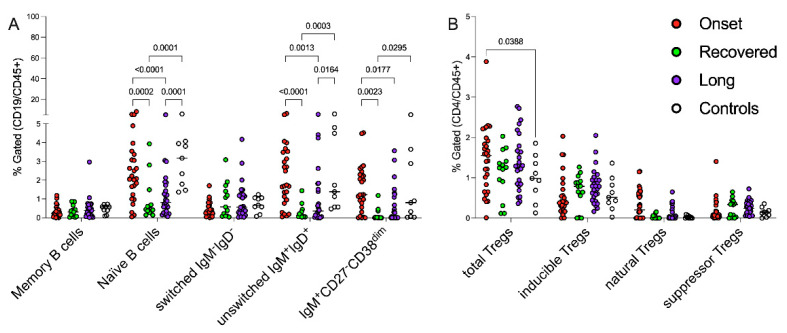
Age-related impact of early/late phases of CoV2 infection on B and Tregs lymphocytes. Gating strategy for the identification of cell subsets is described in Figure 1 and in [21]. Each circle represents the percentage of each cell subset or ratio of the different patients/healthy subjects based on the indicated colors: *CoV2*-infected children at onset of the disease (red circles), who recovered (green circles), with persistent symptoms (violet circles) and healthy subjects (white circles). The panel (**A**) displays the distributions of B-cell subpopulations in the cohort of patients/controls older than 6 years; the panel (**B**) shows the percentages of Tregs subsets in children >6 years old. The data were statistically examined using two-way ANOVA corrected with Tukey.

**Figure 4 jcm-11-04363-f004:**
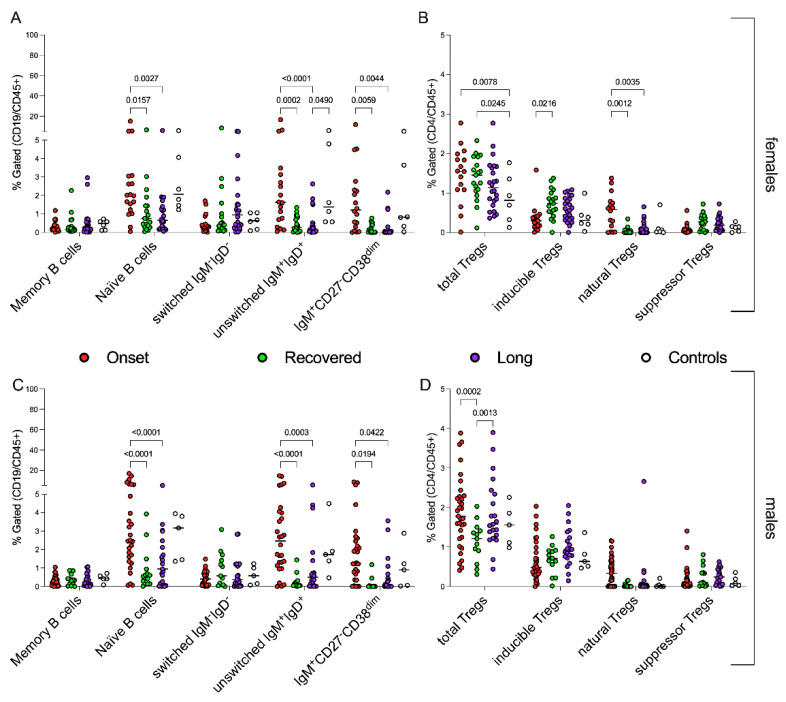
Sex-related B and Tregs subsets, during early/late phases of *CoV2* infection. Each circle indicates the percentage of each cell subpopulation or ratio of the different patients/healthy subjects based on the displayed colors: *CoV2*-infected children at onset of the disease (red circles), who recovered (green circles), with persistent symptoms (violet circles) and healthy subjects (white circles). The panels (**A**,**B**) display the sex-related distributions of B-cell subpopulations; the panels (**C**,**D**) show the percentages of Tregs subsets in the different groups comparing males with females. Statistical analyses were performed with a two-way ANOVA corrected with Tukey.

**Figure 5 jcm-11-04363-f005:**
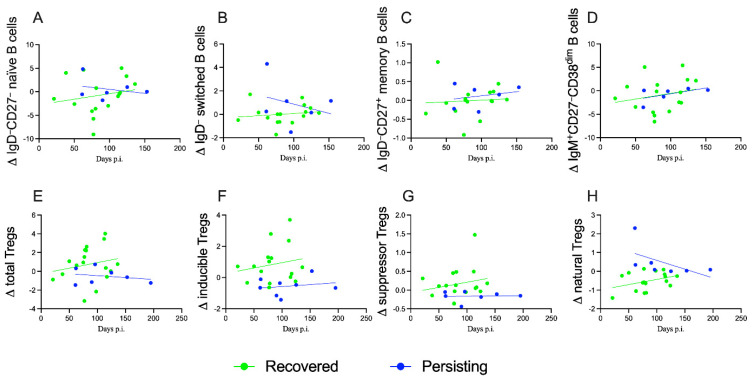
Modulation of T and B cells subsets during follow up of *CoV2-*infected children. Each graph shows the deltas of each variable during time (indicated as days post infection/diagnosis); symbols represent the different patients and are displayed with interpolation lines and confidence bands. The panels compare *CoV2*-infected children who recovered (green circles and lines) with the ones with persistent symptoms (blue circles and lines). (**A**–**D**) display B cells subsets: naïve (F = 0.5453. DFn = 1, DFd = 18 and *p* = 0.4698), switched (F = 1.150. DFn = 1, DFd = 18 and *p* = 0.2977), memory (F = 0.04023. DFn = 1, DFd = 18 and *p* = 0.8433) and pre-transitional (F = 0.000019. DFn = 1, DFd = 18 and *p* = 0.9965) B lymphocytes, respectively. (**E**–**H**) show regulatory T cell subpopulations: total (F = 0.6585. DFn = 1, DFd = 20 and *p* = 0.4266), inducible (F = 0.1222. DFn = 1, DFd = 20 and *p* = 0.7303), suppressor (F = 0.3908. DFn = 1, DFd = 20 and *p* = 0.5389) and natural (F = 5.761. DFn = 1, DFd = 20 and *p* = 0.0262) Tregs, respectively. Simple linear regression revealed significant differences of the slopes only for natural Tregs (**H**).

**Table 1 jcm-11-04363-t001:** Study population. Demographic, clinical and laboratory data of patients are divided into four groups: *CoV2*^+^-infected children with persistent symptoms (Long *CoV2*^+^), at onset (*CoV2*^+^ onset), after recovery of the symptoms (*CoV2*^+^ recovered) and *CoV2*^−^ children. Infected children are subdivided into asymptomatic, with mild and with moderate/severe disease. Patients with severe disease are treated, according to our local protocols, with iv steroids and oxygen support. Patients with mild/moderate disease receive only supportive treatment (iv fluids and antipyretics/pain control).

		Long *CoV2^+^*	*CoV2^+^* Onset	*CoV2^+^* Recovered	*CoV2^−^* Controls
		*n* = 51	*n* = 49	*n* = 32	*n* = 9
**Age**		10.6 ± 4.62	7.1 ± 5.6	7.75 ± 5.5	7.3 ± 5.9
**Females**		26 (51)	17 (34.7)	20 (62.5)	3 (33.33)
Distance from acute infection (months)		4.2 ± 2.16	0	4.2 ± 2.04	0
Severity of Acute Disease	Asymptomatic	4 (7.8)	6 (12.2%)	4 (12.5)	/
	Mild	42 (82.4)	27 (55.1)	26 (81.25)	
	Moderate	5 (9.8)	10 (20.5)	0	
	Severe	0	6 (12.2)	2 (6.25)	
Persistent symptoms in Long COVID children	Headache	19 (37.2)	/	/	/
	*Dyspnea after efforts*	*15 (29.4)*			
	*Muscle-skeletal pain*	*12 (23.5)*			
	Gastrointestinal issues	11 (21.5)			
	Fatigue	10 (19.6)			
	Chest pain	10 (19.6)			
	Tachycardia	6 (11.7)			
	Anosmia	5 (9.8)			
	Dysgeusia	5 (9.8)			
	Skin rashes	5 (9.8)			
	Low grade fever	4 (7.8)			
	Sleeping problems	4 (7.8)			

**Table 2 jcm-11-04363-t002:** B and Tregs populations comparisons. The data refers to Figure 1 extracted percentages (mean ± standard deviation), excluding controls, and *p* values were calculated applying two-way ANOVA corrected with Tukey: a: comparison among Onset and recovered children; b: comparison among onset and long; and c: comparison among long and recovered.

Lymphocyte Populations	Onset	Recovered	Long	*p*-Values		
				a	b	c
IgD^−^CD27^+^ memory B cells	0.3 ± 0.3	0.4 ± 0.4	0.4 ± 0.6	1.0	1.0	1.0
IgD^+^CD27^−^ naïve B cells	3.9 ± 3.9	1.1 ± 1.4	1.2 ± 1.3	<0.0001	<0.0001	1.0
IgD^−^IgM^−^ B cells	0.5 ± 0.4	1.0 ± 1.6	1.0 ± 1.3	0.6	0.5	1.0
IgD^+^IgM^+^ B cells	3.4 ± 6.4	0.4 ± 0.4	0.7 ± 1.2	<0.0001	<0.0001	1.0
IgM^+^CD27^−^CD38^dim^ B cells	2.2 ± 2.5	0.2 ± 0.3	0.4 ± 0.8	<0.0001	<0.0001	0.8
CD25^high^FOXP3^+^ Treg	1.9 ± 0.8	1.3 ± 0.5	1.5 ± 0.8	<0.0001	0.0004	0.14
Inducible Tregs	0.5 ± 0.5	0.7 ± 0.4	0.8 ± 0.5	0.5	0.1	0.9
Natural Tregs	0.4 ± 0.4	0.04 ± 0.07	0.1 ± 0.4	0.0005	0.0026	0.8
Suppressor T regs	0.2 ± 0.3	0.3 ± 0.2	0.2 ± 0.2	0.7	0.8	0.9

**Table 3 jcm-11-04363-t003:** B and Tregs populations in children older than 6 years. Data refer to Figure 3 extracted percentages (mean ± standard deviation), excluding controls, and *p* values were calculated applying two-way ANOVA corrected with Tukey: a: comparison among Onset and recovered children; b: comparison among onset and long; and c: comparison among long and recovered.

Lymphocyte Populations	Onset	Recovered	Long	*p*-Values		
				a	b	c
IgD^−^CD27^+^ memory B cells	0.3 ± 0.3	0.3 ± 0.3	0.5 ± 0.6	1.0	1.0	1.0
IgD^+^CD27^−^ naïve B cells	2.6 ± 1.9	0.9 ± 1.2	1.2 ± 1.2	0.0002	<0.0001	0.9
IgD^−^IgM^−^ B cells	0.4 ± 0.4	0.9 ± 0.9	0.9 ± 0.9	0.6	0.4	1.0
IgD^+^IgM^+^ B cells	2.2 ± 1.7	0.3 ± 0.4	1.0 ± 1.5	<0.0001	0.001	0.2
IgM^+^CD27^−^CD38^dim^ B cells	1.6 ± 1.3	0.2 ± 0.3	0.6 ± 0.9	0.002	0.02	0.7
CD25^high^FOXP3^+^ total Treg	1.5 ± 0.8	1.1 ± 0.6	1.4 ± 0.7	0.1	1.0	0.3
Inducible Tregs	0.5 ± 0.5	0.7 ± 0.4	0.8 ± 0.4	0.7	0.2	0.9
Natural Tregs	0.3 ± 0.4	0.03 ± 0.05	0.1 ± 0.2	0.2	0.3	1.0
Suppressor T regs	0.2 ± 0.3	0.3 ± 0.2	0.3 ± 0.2	0.9	0.9	1.0

## Data Availability

Data are available upon request to the corresponding author.

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
