# Peer review of "Recovering or Persisting: The Immunopathological Features of SARS-CoV-2 Infection in Children"

_jcm, 2022, doi:10.3390/jcm11154363_

Round 1
Reviewer 1 Report
The article assesses serum cytokines and T and B cell subsets in children during SARS-CoV-2 and post-infection.
The topic is timely and interesting. The article is well-written and clear.
I have some comments:
- the number of controls is limited and, among them, high variability is observed in the presented analyses. Is it possible to increase the n of the controls? Moreover, a control matching for age and sex should be attempted to validate the results.
- Regarding T cells, only Treg are shown and discussed. Is it possible to show T cell subsets (naive, effector memory,...)?
- Figure 2: "Circles represent the percentage of each cell population/subset or ratio of the single patients/healthy subjects comparing the four groups based on cytokine expression levels". However, the unit is pg/ml. Could you please specify?
- Figure 2B: could you please specify why only 4 controls are shown?
Author Response
The article assesses serum cytokines and T and B cell subsets in children during SARS-CoV-2 and post-infection.
The topic is timely and interesting. The article is well-written and clear.
I have some comments:
- the number of controls is limited and, among them, high variability is observed in the presented analyses. Is it possible to increase the n of the controls? Moreover, a control matching for age and sex should be attempted to validate the results.
In the study design we focused mainly on the comparison of children affected by long covid and children who fully recovered. All the patients were enrolled among children who were infected during the first and second waves of the pandemic. In these phases, for the strong restrictions in Italy, it was very difficult to enroll healthy children admitted to the hospital for non-infectious and not-immuno-mediated diseases. Successively, after the analysis of the results obtained, we were conscious that we enrolled a limited number of controls and that among them there were a certain heterogeneity. On the other hand, however, we excluded to enroll new healthy children, because it is currently very difficult to find a child in Italy who has never been in contact with either Sars-CoV-2 infection or who has not been vaccinated. This aspect could generate a further bias or element of heterogeneity. For this reason, and we thank the reviewer for having noticed, we decided to not focus our manuscript on the significant differences between infected groups and the healthy one, highlighting the comparisons among infected children.
- Regarding T cells, only Treg are shown and discussed. Is it possible to show T cell subsets (naive, effector memory,...)?
The hypothesis of our work was based on the analysis of the potential imbalance of regulatory ability and B cell subpopulation of immune system during different phases of Sars-CoV-2 infection, focusing for T cells only on regulatory subsets. We did not deepen other T cell compartments, equally of interest, but for which more in-depth testing such as antigen specificity would have been needed.
- Figure 2: "Circles represent the percentage of each cell population/subset or ratio of the single patients/healthy subjects comparing the four groups based on cytokine expression levels". However, the unit is pg/ml. Could you please specify?
We changed the figure legend as following:
Each Circle represents the concentration (expressed as picograms/milliliter, pg/ml) of the cytokines for each patient/healthy subject and the 4 plots compare for each cytokine the four groups.
- Figure 2B: could you please specify why only 4 controls are shown?
IL8 plot (Figure 2B) displays a limited number of patients/healthy subjects. The missing ones are not showed because they resulted undetectable. We explained this aspect in the emendated figure legend.
Reviewer 2 Report
Dear authors thank you for the opportunity to review this article. It is very interesting and I believe it is an important contribution to our knowledge of LOng-COVID syndrome. However, I would like to ask for some clarifications/additions
1) In the introduction, I am missing a paragraph that describes our knowledge of long-COVID in children. What is its frequency? Risk factors, etc.
2) The introduction should also include the definition of long-COVID that the authors use in the manuscript. Admittedly, the authors mention it in part in the methodology, but there is no indicated source and no explanation of why this and not another definition was chosen.
3) How was the control group selected? The authors do not mention this in the methodological description. Why were only 9 children included in the control group?
4) In the limits of the work, I miss the mention that we do not know the baseline condition of the patients, whether they have chronic diseases, whether and what medications they take chronically. How were they treated during the acute phase of COVID-19.
5) In the discussion, it is still worth adding that a specific group of patients may respond differently to the disease/vaccination, especially children with immune deficiencies. (https://doi.org/10.3390/jcm10215111)
Overall, the article is an important voice on long-COVID in children, about which we know very little.
Author Response
Dear authors thank you for the opportunity to review this article. It is very interesting and I believe it is an important contribution to our knowledge of LOng-COVID syndrome. However, I would like to ask for some clarifications/additions
1) In the introduction, I am missing a paragraph that describes our knowledge of long-COVID in children. What is its frequency? Risk factors, etc.
We added a paragraph who dissect this aspect: “So far, there is not yet an internationally agreed definition of pediatric PCC. According to the NICE guidelines, the term ‘long COVID’ is commonly used to describe signs and symptoms that continue or develop after acute COVID 19. It includes both ongoing symptomatic COVID 19 (from 4 to 12 weeks) and post COVID 19 syndrome (12 weeks or more) (https://www.nice.org.uk/guidance/ng188/resources/covid19-rapid-guideline-managing-the-longterm-effects-of-covid19-pdf-51035515742 accessed 19/07/2022). Conversely, a more recent Delphi process proposed a new definition of PCC, being the persistence of “at least one persisting physical symptom for a mini-mum duration of 12 weeks after initial testing that cannot be explained by an alternative diagnosis. The symptoms have an impact on everyday functioning, may continue or develop after COVID infection, and may fluctuate or relapse over time. [14]”. For these reasons, data about its prevalence are affected by the definitions used and the included populations, ranging from 1 to 10% of cases, with older age and allergy being the main risk factors [15].”
2) The introduction should also include the definition of long-COVID that the authors use in the manuscript. Admittedly, the authors mention it in part in the methodology, but there is no indicated source and no explanation of why this and not another definition was chosen.
This is an intriguing aspect. Currently the definition of Long-COVID is still debated. However, we added some clarification also in the introduction and methods clarifying which definition we used and why.
3) How was the control group selected? The authors do not mention this in the methodological description. Why were only 9 children included in the control group?
In the study design we focused mainly on the comparison of children affected by long covid and children who fully recovered. All the patients were enrolled among children who were infected during the first and second waves of the pandemic. In these phases, for the strong restrictions in Italy, it was very difficult to enroll healthy children admitted to the hospital for non-infectious and not-immuno-mediated diseases. Successively, after the analysis of the results obtained, we were conscious that we enrolled a limited number of controls and that among them there were a certain heterogeneity. On the other hand, however, we excluded to enroll new healthy children, because it is currently very difficult to find a child in Italy who has never been in contact with either Sars-CoV-2 infection or who has not been vaccinated. This aspect could generate a further bias or element of heterogeneity. For this reason, and we thank the reviewer for having noticed, we decided to not focus our manuscript on the significant differences between infected groups and the healthy one, highlighting the comparisons among infected children.
4) In the limits of the work, I miss the mention that we do not know the baseline condition of the patients, whether they have chronic diseases, whether and what medications they take chronically. How were they treated during the acute phase of COVID-19.
Thank you very much, we further clarified these important points. Specifically, we added that “Patients with confirmed or suspected primary or acquired immune compromising conditions, recent or current administration of immune suppressive therapies, or other diseases affecting the immune system, or patients with chronic comorbidities and genetic disorders that might affect immune responses or make difficult the recognition of subtle symptoms of long covid were excluded from the study”. Therefore, those undergoing chronic medications were excluded.
Regarding treatments during the acute phase, we clarified in the table 1 legend what they received in case of severe or non-severe disease: “Patients with severe disease are treated, according to our local protocols, with iv steroids and oxygen support. Patients with mild/moderate disease receive only supportive treatment (iv fluids and antipyretics/pain control)”
5) In the discussion, it is still worth adding that a specific group of patients may respond differently to the disease/vaccination, especially children with immune deficiencies. (https://doi.org/10.3390/jcm10215111)
Thank you, this is an important point to clarify. We have added the following: It is important to highlight that specific pediatric populations, including those immune deficiencies, may present specific characteristics in post-acute immune responses, as demonstrated during acute infection or vaccination [62,63]. In fact, as we have excluded these populations, our results may not be directly translated in children with diseases affecting the immune system.
Overall, the article is an important voice on long-COVID in children, about which we know very little.
Thanks a lot